# Attention-Based Deep Multiple-Instance Learning for Classifying Circular RNA and Other Long Non-Coding RNA

**DOI:** 10.3390/genes12122018

**Published:** 2021-12-19

**Authors:** Yunhe Liu, Qiqing Fu, Xueqing Peng, Chaoyu Zhu, Gang Liu, Lei Liu

**Affiliations:** 1Institute of Biomedical Sciences, Fudan University, Shanghai 200433, China; yunhe_liu15@fudan.edu.cn (Y.L.); 19111010045@fudan.edu.cn (Q.F.); 18111510058@fudan.edu.cn (X.P.); 18111510028@fudan.edu.cn (C.Z.); 2School of Basic Medical Science, Fudan University, Shanghai 200433, China

**Keywords:** non-coding RNA, circRNA, deep learning, MIL architecture, sequence motif

## Abstract

Circular RNA (circRNA) is a distinguishable circular formed long non-coding RNA (lncRNA), which has specific roles in transcriptional regulation, multiple biological processes. The identification of circRNA from other lncRNA is necessary for relevant research. In this study, we designed attention-based multi-instance learning (MIL) network architecture fed with a raw sequence, to learn the sparse features of RNA sequences and to accomplish the circRNAs identification task. The model outperformed the state-of-art models. Moreover, following the validation of the attention mechanism effectiveness by the handwritten digit dataset, the key sequence loci underlying circRNA’s recognition were obtained based on the corresponding attention score. Then, motif enrichment analysis identified some of the key motifs for circRNA formation. In conclusion, we designed deep learning network architecture suitable for learning gene sequences with sparse features and implemented it for the circRNA identification task, and the model has strong representation capability in the indication of some key loci.

## 1. Introduction

Non-coding RNAs (ncRNAs), referring to RNAs without protein-coding potential, account for the majority of RNAs. It is generally recognized that lncRNA (long non-coding RNA) is a kind of ncRNAs longer than 200 nucleotides, which distinguishes itself from other smaller ncRNA species such as miRNAs and siRNAs. lncRNA has complex biological functions such as transcriptional regulation and post-transcriptional control [1,2,3]. Circular RNA (circRNA) is a closed formed lncRNA by covalently closed loops. Based on current researches, circRNAs are more stable than mRNAs and play a major role as a microRNA activity modulator. CircRNAs are also relevant to the development of multiple diseases [3,4,5], and can be used for disease biomarkers [6,7]. Therefore, it is vital to detect circular RNAs.

Currently, some computational approaches in identifying circRNA [8,9,10] have been developed with different frameworks. For example, CirRNAPL [11] adopted the extreme learning machine based on the particle swarm optimization algorithm. CircLGB utilized a LightGBM classifier to categorize circRNA [12]. Based on the deep learning framework, circDeep [13] utilized a fused structure (RCM (reverse complement matching) descriptor, asymmetric CNN-BLSTM descriptor, and sequence conservation descriptor) to achieve higher identification accuracy compared with exiting tools.

For these models mentioned above, the input was not the raw sequence, but often the relevant features extracted from the predicted secondary structure [9,13]. For circDeep, a deep learning framework incorporated a complementary score [14] and a conservation score of the sequence. Its sequence input part did not use the full-length RNA sequence and underwent a triplet transformation [15], either. It is important to find a deep learning framework suitable for sequence input as well as sequence learning, to facilitate the utilization of algorithms and take advantage of the information in sequence.

The characteristics of RNA sequences are quite different from the other sequence data such as word language. We organized the differences into three main points. First, the RNA sequence is a combination of multiple meaningful and meaningless units, where the meaningful units are embedded into the entire background sequences, not like the words formed by a certain grammatical structure [16]. An RNA typically has a large variety of functions enabled by meaningful units, such as the ability to form high-level structures and to recruit other components [17]. While learning models tend to have singular learning objectives, such as distinguishing circular RNA, which results in the meaningful units for the learning model is sparse among the full-length sequence. Second, the length of different RNA varies greatly [18], spanning from 100 to 1,000,000 nt, suggesting that the density of the meaningful units also varies considerably. Third, the character component of the RNA sequence is relatively simple, which only contains four characters (ATGC) and the single character is meaningless. On the other hand, the composition and length of meaningful components are unknown, so the input data for learning can only be the character itself instead of a meaningful word.

To address the problem mentioned above, we designed an attention-based deep encoder MIL (multiple-instance learning) model (Circ-ATTEN-MIL). The MIL structure is suitable for learning sparse features [19,20], and the attention-based pooling layer can discover similarities among instances and has a stronger representation capability [21]. We applied this deep network structure to learn the identification task, which achieved better accuracy, and extracted high attention sequences to enrich motifs, which shed light on studies regarding RNA circligase.

## 2. Materials and Methods

### 2.1. Data Source

CircRNAs sequences were extracted from the circRNADb database [22] and other lncRNAs sequences were extracted from the GENCODE database [23] (lincRNA, antisense, processed transcript, sense intronic, and sense overlapping), respectively. After removing sequences shorter than 200 nucleotides, we got 31,939 circRNAs and 19,722 other lncRNAs. The circRNA sequences were regarded as positive samples. We randomly divided the dataset into a training set (75%), validation set (10%), and test set (15%).

### 2.2. Instances Extraction by Sliding Window

An RNA sequence was regarded as a bag, and instances were extracted from the sequence. For each full-length sequence, we connected the head (5′ end) and tail (3′ end) of the sequence, set the slider window size and the sliding step, and made the slider move from the head. For each step, the sequence contained in the window was extracted as an instance, until the slider moved out of the tail of the sequence (illustrated in Figure 1). For a sequence of a certain length, the number of instances can be calculated by the following formula.
(1)$$\mathrm{Number}\;\left(\mathrm{Instances}\right)={\left[\;\frac{\mathrm{Length}\;\left(\mathrm{Sequence}\right)}{\mathrm{Step}\;\left(\mathrm{Slide}\right)}\;\right]}_{round}$$

### 2.3. Model Structure

The network structure is represented in Figure 2. We employed the encoder structure of the seq2seq model [24] here as the instance feature extractor. The embedding layer [25] was employed to represent bases (15 (A, T, G, C, N, H, B, D, V, R, M, S, W, Y, K)→4 (representative dimension)). The encoder used a bi-directional RNN structure, which gave equal attention to the head and the tail of the instance, and the output was a context vector [26] to represent the feature of the instance. And subsequently, through the MIL layer, the features of all instances were scored and aggregated jointly to determine the type of the bag [20,21,27].

### 2.4. Attention Mechanism as the MIL Pooling

Referring to previous work on the pooling layer structure, we selected the attention-based pooling structure, which exhibited better aggregation and representation capacity [21]. It was assumed that the features extracted by the encoder were $C=\{c_{1},\;\ldots ,\;c_{k}\}$ (The dimension of each $c_{k}$ is M), and its corresponding attention weights were $\alpha =\{\alpha_{1},\;\ldots ,\;\alpha_{k}\}$, which could be formulated as follows.
(2)$$\alpha =softmax\left(W^{T}\tan h\left(VC^{T}\right)\right)$$
where $W\in R^{L\times 1}$ and $V\in R^{L\times M}$ are the parameters of the two network layers connects the feature (C) to the attention score. The attention-based structure allowed us to discover the similarity between different instances and made the network have better representability. After the encoder feature was weighted by the attention scores, the probability of determination was output via a sigmoid neuron through a fully connected layer.

### 2.5. Handling of Handwritten Numbers Dataset

The handwritten numbers dataset was used to verify the representational power of the attention score. Each number figure (size = 28 × 28) was served as an instance. A bag contained more than 16 instances. For each instance, we treated the image as a sequence containing 28 characters, and each with a representation dimension of 28, for feeding into the network (Circ-ATTEN-MIL; the embedding layer in encoder block was removed in this task) (Figure 3). A bag is positive when it contains the determining number (two modes were set: determining number is 0; determining numbers are 0, 1, 3).

### 2.6. Fusion Model

The ‘weighted feature’ (the penultimate layer) of Circ-ATTEN-MIL was extracted as the sequence feature. The other features were calculated using the extraction methods of RCM descriptor and sequence conservation descriptor in CircDeep. Combining these three types of features (sequence feature: 100; RCM feature: 40; conservation feature: 23), a four-layer MLP (multi-layer perceptron) network (163-80-20-1 (the output layer is a sigmoid-activated neuron)) was constructed as a fusion model.

### 2.7. Evaluation Criteria

We evaluated the model performance by classification accuracy, sensitivity, specificity, MCC (Matthew’s correlation coefficient), and F1 score (formulated as follows).
(3)$$\mathrm{MCC}=\frac{TP\times TN-FP\times FN}{\sqrt{\left(TP+FP\right)\left(TP+FN\right)\left(TN+FP\right)\left(TN+FN\right)}}$$
(4)$$F1=\frac{2\times precision\times recall}{precision+recall}$$

### 2.8. Extraction of High-Attention Sequence Splices

As the attention score was applied to the encoder features of each instance, we assigned the same scores to the sequence of the instance, and collapsed the weighted sequences according to the inverse of the slider rule (Figure 4), and extracted the sequence fragment (with certain length: >7) with the higher attention score (after scaling to between 0 and 1: >0.6), which served as the high-attention sequence splices.

### 2.9. Motif Enrichment

MEME software [28] was utilized to perform motif enrichment tasks. In the MEME environment, classic mode was selected to enrich motifs in RNA sequences between 6 and 50 lengths (the code was: meme RNA.fasta-RNA-nostatus-mod zoops-minw 6-maxw 50-objfun classic-markov_order 0).

## 3. Results

### 3.1. Dataset Description

The sequence length distribution and base proportion between circRNAs and other lncRNAs (In training set) were very similar (Figure 5), which illustrated that the simple features between the two-type sequences were comparable and the model fed with raw sequences was hard to accomplish the identification task by these simple features.

### 3.2. Model Architecture

In instance extraction, the window size was set to 70 and the sliding step was set to 5 (Figure 1). In the encoder block, it consists of one embeding_15_4 layer and two bi-direction LSTM_4_150 layers. The final step outputs of both directions were concatenated, and via an FCN_300_100 (the fully connected network consists of two layers of 300 and 100 neurons in turn) layer, the instance feature (C_100) was obtained. In the attention block, the C_100 features of each instance were accepted as key values. After an FCN_100_30, an FCN_30_1 layer, the dimension for each instance was reduced to 1 (attention value). A softmax layer was utilized to normalize the attention value for each instance, and then the normalized attention score was yielded. Finally, the classifier block accepted all instances’ weighted C_100 feature, through a fully connected layer and a sigmoid neuron, and outputted the identification probabilities (Figure 2).

### 3.3. Model Training and Identification Evaluation

We used the binary cross-entropy loss function to calculate loss and trained the models with the Adam optimization algorithm (the learning rate is 0.0002; betas = (0.9, 0.999); the weight decay is 10^−5^). Balancing the accuracy and over-fitness, we chose the model trained at the 70th epoch as the final model and plotted the ROC curve (Figure 6). As a result, the performance of the model training had strong identification efficiency (train AUC = 0.99; validation AUC = 0.97; test AUC = 0.97). Subsequently, multiple evaluation criteria were employed to test the model (Table 1), and these metrics also validated that the model has a high degree of robustness.

### 3.4. Comparison with Other Algorithms

The Circ-ATTEN-MIL model was compared with a classical circRNA identification model, PredcircRNA [9], and a deep learning architecture-based end-to-end model, CircDeep [13]. The ACNN-BLSTM descriptor in CircDeep took a partial RNA sequence of a set certain length (the default is 8000) as the input and could be served as a separate classifier. While in Circ-ATTEN-MIL, the input consisted of full RNA sequences with variable lengths. The comparison results showed that Circ-ATTEN-MIL was better than the PredcircRNA and ACNN-BLSTM descriptors under the three metrics and almost the same compared to CircDeep (Table 2). Finally, we incorporated the RCM descriptor and the sequence conservation descriptor, which were used as input feature in the CircDeep model beyond RNA sequences, with Circ-ATTEN-MIL to build a fusion model (in the Materials and Methods section), and successfully improved the discriminative power of the final model.

### 3.5. Attention Score Employed for Identifying Determining Factor

To verify the representational power of the attention score, we used the handwritten numbers dataset to visualize the known determining factor with the produced attention score. Two model (in the encoder block: 2 LSTM_28_10, FCN_10_10; in MIL block: FCN_10_5, FCN_5_1) was trained in this part, one (model 1) with 0 and another (model 2) with 0, 1, 3 as decisive factor (a bag containing decisive factors was treated as positive sample). The training was stopped after the accuracy exceeded 0.90 (around 10 epochs). We visualized the attention score with the matched instances and discovered that the attention score works well whether the bag contains a single determinant, multiple identical determinants, or multiple different determinants (Figure 7). Statistics on the decisive factor identification showed a very low percentage of false identifications, and although there was a certain unrecognized rate, the identified numbers had a very high confidence level (>99%).

### 3.6. Motif Enrichment from High-Attention Sequence

The high attention sequences were extracted for all correct identification circRNA transcripts. Most of the high-attention sequences were between 8–40 in length, and the count of the attention sequences for each transcript was around 4 (Figure 8), which validated our initial assumption that the meaningful features were sparse. All high attention sequences were used for motif enrichment, and multiple validated motifs were yield (Table 3).

## 4. Discussion

In this project, we designed a deep learning network architecture suitable for learning gene sequence features and implemented the model to accomplish the circRNA identification task. Based on the attention score produced by the model, a large number of key sequence loci for circRNA recognition were extracted. Following the motif enrichment analysis, some possible key motifs for circRNA formation were identified.

The post-transcriptional modifications and a variety of related functions of transcripts are encoded in their sequence [29]. Thus, a sequence contains a large number of key loci responsible for each of the processes [30]. For machine learning models, which are often required to identify only a single function among these processes, such as loop formation, the entire sequence can be too redundant and the meaningful features are too sparse. From another viewpoint, the learning-by-sequence task is similar to multiple instance learning (MIL) [20,27,31], that is, for weak label learning problems with sparse features. We changed the convolutional blocks commonly used in the MIL related-model for feature extraction to an RNN block that is more suitable for sequence learning [32], and used the attention mechanism [21,33] as the MIL layer, which has stronger representation capability. The results demonstrate the validity of the structure and the great potential value of the attention mechanism.

For this circRNA identification task, data were collected from the validated reference sequence database [22,23], with the attendant problem of low sampling rates. If a single gene is assumed to be a single distribution (which may actually be a set of genes), the use of a reference sequence causes only one sample to be collected for a single distribution and the sampling rate can be considered to be relatively low. If multiple actual sequences can be collected for a single gene, which implies that there may be a variety of mutations in non-relevant features among multiple sequences and the relevant features are more conservative. Therefore, the increased sampling rate must enhance the model’s learning ability and improve its discriminative power. Considering that data collection is more difficult [34], it is worthwhile to explore improving the effectiveness of the model by trying some data augmentation methods.

The instance is extracted by a moving slider, which can only extract the continuous regional features in the sequence. However, sequences form higher level stereo structures in space [35], so the key feature can be the combinations of sequences that are far apart. Considering this possibility, adding more mechanisms for instances extraction and combination, to make a single instance can contain multiple combinations of distant sequences, may further improve the discriminative effectiveness as well as the potential representational value of this network structure.

The model can be used for more than just the identification of circRNAs. Since only the original sequence is required as input, the network structure can be used for learning other sequence-related tasks by simply changing the resultant events. Because of its representation capability, it can be used to discover key sequences for different tasks and provide a basis for other relevant research.

## 5. Conclusions

Circ-ATTEN-MIL was designed and used for circRNA identification, and it outperformed other deep learning models currently used. The model utilized the MIL-attention network architecture, which took the complete RNA sequence as input and not only carried out the discriminant probability of circRNA, but also outputted the score of the importance of each instance, which could be used for identifying the critical part of a sequence for model judgment and would be able to provide some insights for basic research in related fields.

## Figures and Tables

**Figure 1 genes-12-02018-f001:**
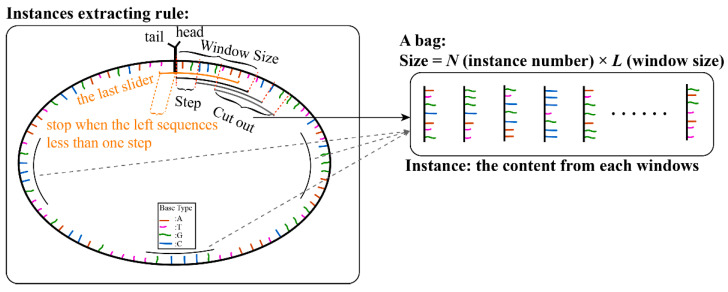
Illustrations of instance extraction from full RNA sequence.

**Figure 2 genes-12-02018-f002:**
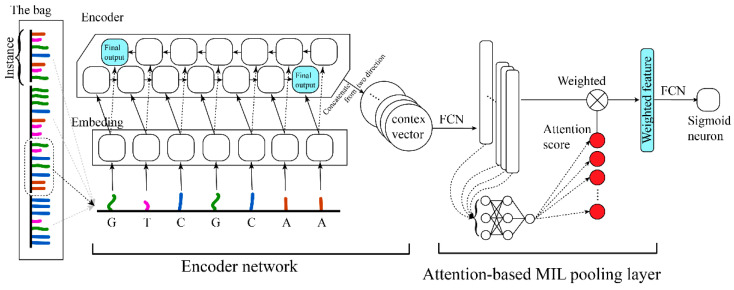
Illustrations of attention-based deep encoder MIL model structure (Circ-ATTEN-MIL).

**Figure 3 genes-12-02018-f003:**
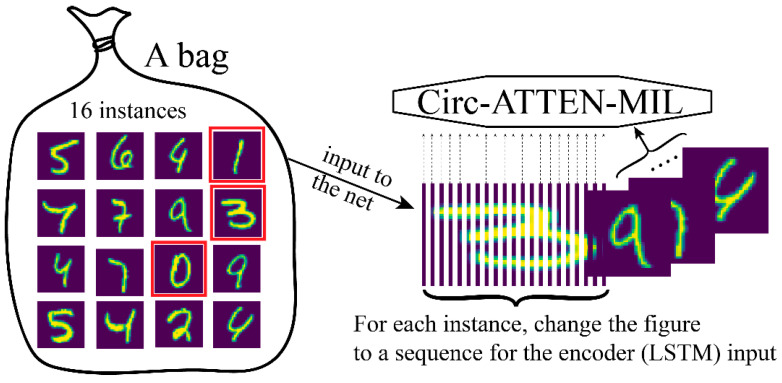
Handling of handwritten numbers dataset for feeding into Circ-ATTEN-MIL.

**Figure 4 genes-12-02018-f004:**
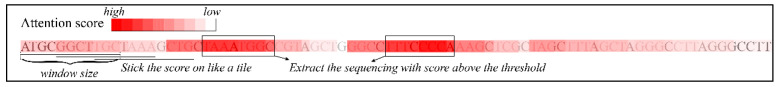
Illustrations of extraction of high-attention sequence splices.

**Figure 5 genes-12-02018-f005:**
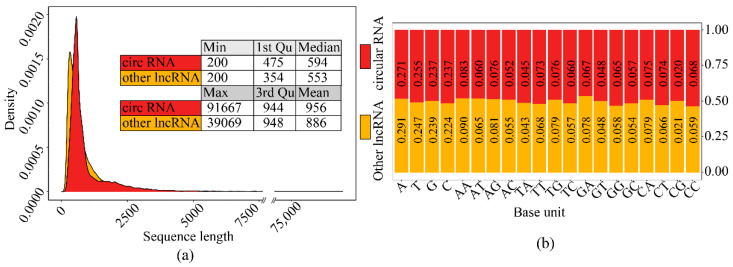
The comparison of simple features of sequences between the two-type sequence set: (**a**) Sequence length distribution comparison. (**b**) Sequence composition comparison.

**Figure 6 genes-12-02018-f006:**
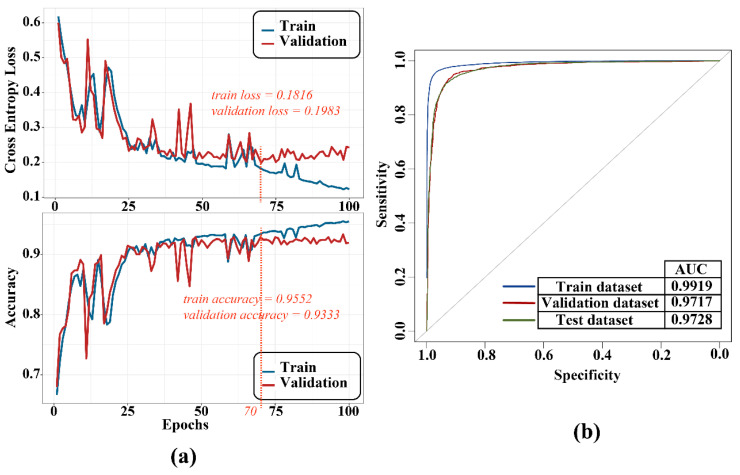
Training process (**a**) and ROC curve (**b**).

**Figure 7 genes-12-02018-f007:**
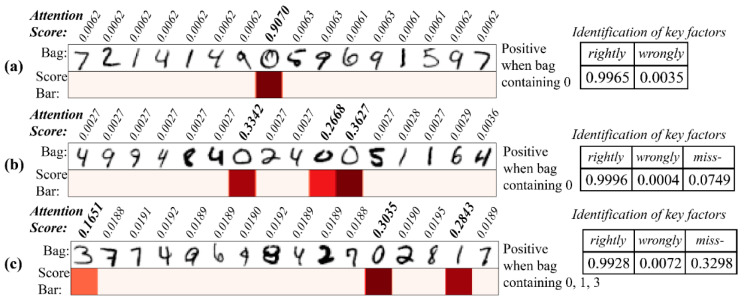
Attention score for identifying the determining numbers. (**a**) Single determinant (model 1); (**b**) multiple identical determinants (model 1); (**c**) multiple different determinants (model 2). (Left panel: attention score bar; right panel: the rightly and wrongly identify events and miss-identify events.)

**Figure 8 genes-12-02018-f008:**
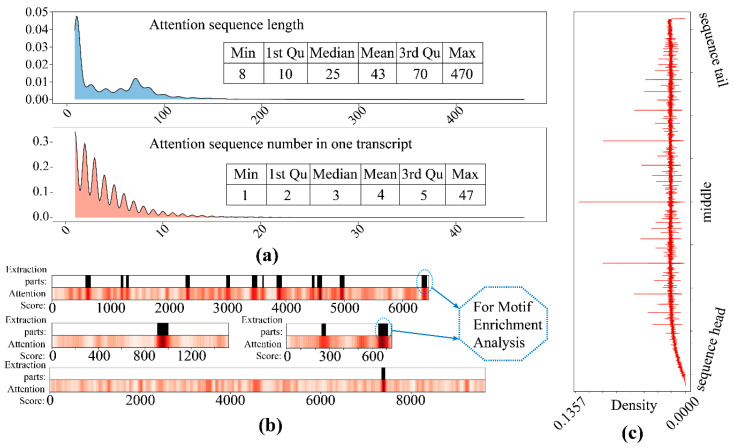
The high attention sequence distribution. (**a**) The length distribution (upper) and the attention sequence number for each transcript distribution (lower); (**b**) the extraction of attention sequence for motif enrichment; (**c**) density distribution of attention loci on all sequences.

**Table 1 genes-12-02018-t001:** The evaluation for classification task.

	Accuracy	Sensitivity	Specificity	Precision	MCC	F1
Train	0.9552	0.9662	0.9547	0.9713	0.9194	0.9687
Validation	0.9333	0.9485	0.9092	0.9433	0.8291	0.9459
Test	0.9284	0.9396	0.9039	0.9393	0.8435	0.9394

**Table 2 genes-12-02018-t002:** The comparison results.

	ACC	MCC	F1 Score
PredcircRNA	0.8056	0.6113	0.8108
ACNN-BLSTM	0.8942	0.7756	0.9149
Circ-ATTEN-MIL	0.9284	0.8435	0.9394
CircDeep (fusion)	0.9327	0.8536	0.9304
Fusion model	0.9434	0.8796	0.9546

**Table 3 genes-12-02018-t003:** Motif enriched from the sequence.

Motif	Sequence	E-Value	Predicted (Uppercase in the Sequences: Target Loci)
** 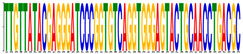 **	tTGTTATACGAGGGATC CCGGTGTCAGGTGGGA GTACTGCAACCTGacgc	2.3 × 10^−9^	KR super family (autonomous structural domains): Kringle domains are believed to play a role in binding mediators. (Source: NCBI)
** 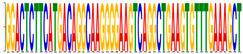 **	ggactcttcatgacAGGC AAGGGGAAGTCAG GCTgaagtgtttgaaagct	2.1 × 10^−3^	SPI1 target sequence: May bind RNA and modulate unmatured-RNA splicing by similarity. (Source: JASPAR)
** 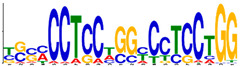 **	yssmccTCCWGGYCC TCCTGG	6.7 × 10^4^	PLN02915 super family: catalytic subunit. (Source: NCBI)Contains ETS1 target sequence. (Source: JASPAR)
** 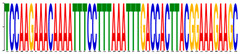 **	tCCAAGAAACAAAAT TTCCTTTAAATTTGAC CACTTACGGAAAGAAgc	2.0 × 10^5^	The actual alignment was detected with superfamily member pfam01267. (Source: NCBI)
** 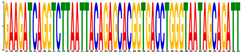 **	gaagatcaggtcttaATTA CAGAGCACGGTGA CCTGggtaatagcagatt	1.1 × 10^6^	Contains multi estrogen receptor (ESR1; ESR2) and estrogen related receptor (ESRRA) target sequences. (Source: JASPAR and GeneCardsSuite)
** 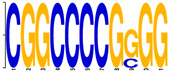 **	CGGCCCCGGGG	2.1 × 10^8^	TFAP2E target sequence: may bind to the consensus sequence 5’-GCCNNNGGC-3’. (Source: JASPAR)
** 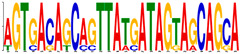 **	agtgacaGCAGTTAT GATagtagcagca	2.8 × 10^7^	Contains multi-Homebox-related factor (A6, A4, B6, C10, C8, D8, B9, B8, B6, B3, A5, A7, A9, B4, C4, A6) target sequences. (Source: JASPAR and GeneCardsSuite)

## Data Availability

The data and code are available in https://github.com/liuyunho/Circ-ATTEN-MIL (accessed on 16 December 2021).

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
