# Peer review of "Attention-Based Deep Multiple-Instance Learning for Classifying Circular RNA and Other Long Non-Coding RNA"

_genes, 2021, doi:10.3390/genes12122018_

Round 1
Reviewer 1 Report
Liu et al. designed an attention-based multi-instance learning (MIL) network for gene sequence to identify circRNA. The whole study is complete in its current form, and I will suggest accepting with minor edits.
I will suggest that the authors add a detailed description of the files and steps to repeat the analysis in the GitHub code repository.
Author Response
Response to Editor and Reviewer 1:
Dear Editor and Reviewer:
We would like to thank genes for giving us the opportunity to revise our manuscript and the reviewers for their careful read and thoughtful comments on the previous draft.
We have carefully taken their comments into consideration in preparing our revision. The following summarizes how we responded to reviewer 1 comments.
Thanks again.
Best wishes.
Revision-authors’ response:
Thank you very much for your recognition of our work and suggestions on the code description on gitup. In this revised version, we have further improved the overall English description and paragraph logic of the article to make it easier to read and understand.
Regarding the code, we are working on packaging it into easy-to-use software. In the current version, we just publish the detailed structure and running parameters in the article and the key code block in the gitup. And we will publish the packaged software in the near future, which can be used not only for circRNA identification, but also as a learning tool for different target RNA sequences, and can give the key sequences directly after the training process.
Thank you again for your recognition of our work, and we hope we can in the future develop more related tools based on this foundation.

Reviewer 2 Report
The authors propose a novel deep learning approach for circRNA identification. The main improvement wrt the existing solutions lies in the application of attention-based multiple instance learning approach. The authors show that their framework outperforms recent deep learning classifiers such as circDeep. Moreover, the individual instances and attention can help to identify circRNA motifs, the key parts for circRNA identification. These features are clear strengths of the paper.
On the other hand, the main weakness is in presentation. Language is poor and paper definitely needs THOROUGH English proofreading. Moreover, the authors should be much more careful in explaining the basic principles and, last but not least, the abbreviations. For example, they simply state that circDeep fuses RCM, ACNN-BLSTM without any explanation what these terms mean. As the authors further work with these abbreviations, the paper is not self-contained then. FCN is other abbreviation without explanation. The meaning of W, V , L and M should clearly be mentioned too.
As for more principled omissions, I would strongly recommend better MIL motivation in circRNA domain. The explanation with MNIST task is nice, however, clear biological motivation for circRNA motifs would be helpful.
In the result section, there is no reference to PredictRNA (taken probably form [9]). The nature of this method is not touched too. The comparison with circDeep and with its different extensions and combinations is done in the text, but it is very brief. For example, a picture that shows the principled changes among ACNN-BLSTM, Circ-ATTEN-MIL, CircDeep and Fusion model could help the reader easily understand benefits of the individual components of the eventual system. Last but ot least, it is not clear why the authors present results in a less detailed form than in previous section 3.3. (sensitivity, specificity, precision).
In the motif discovery part, I would appreciate a better explanation of the relationship between instances (70nt) and high attention sequences (8-40nt).
To conclude, the proposed framework is very interesting, but a major revision of the text is definitely needed.
Author Response
Response to Editor and Reviewer 2:
Dear Editor and Reviewer:
We would like to thank genes for giving us the opportunity to revise our manuscript and the reviewers for their careful read and thoughtful comments on the previous draft.
We have carefully taken their comments into consideration in preparing our revision. The following summarizes how we responded to reviewer 2 comments.
Thanks again.
Best wishes.
Revision-authors’ response:
Thank you very much for your recognition of our work and the suggestions. Regarding the language problem of the manuscript, we thoroughly and carefully checked the full manuscript and corrected several grammatical errors, and changed the logic of some sentences to make the manuscript easier to read and understand. In response to your remaining questions, we have made specific changes for each part of the questions, and give a detailed explanation of each part as below.
Question 1: The authors should be much more careful in explaining the basic principles and, last but not least, the abbreviations. For example, they simply state that circDeep fuses RCM, ACNN-BLSTM without any explanation what these terms mean. As the authors further work with these abbreviations, the paper is not self-contained then. FCN is other abbreviation without explanation. The meaning of W, V , L and M should clearly be mentioned too.
Response: Thank you very much for noticing these weaknesses in our manuscript. For the abbreviation RCM, ACNN-BLSTM problem, we added its full name and easier-to-understand writing style in lines 37-40 of the introduction section, and we improved the logic of this sentence to make a better understanding of the description of the circDeep model. For the abbreviation of FCN, we gave a detailed description in Result 3.2 where the term first appears (lines 161-162). For the meaning of W, V, L, and M in equation 2, We gave a detailed explanation in lines 102 and 105-106. In addition, we checked some other unclear descriptions in the manuscript and correct them accordingly. Thanks very much for this question, and we have made the changes accordingly to make the article description more complete.
Question 2: As for more principled omissions, I would strongly recommend better MIL motivation in circRNA domain. The explanation with MNIST task is nice, however, clear biological motivation for circRNA motifs would be helpful.
Response: Thank you very much for asking this question. The MIL (Multi-Instance Learning) structure is chosen because in the field of picture recognition, for example, the recognition of clinical case pictures, the proportion of lesions in the whole picture is very low. And by analogy with the sparse meaningful features of RNA sequences, we believe that the two are similar and therefore choose to modify the traditional MIL-related network structure for the task of our study. We have added more description about this in lines 64-66 in the introduction, and 238-243 in the discussion section. Thank you for asking this question, and we hope that the changes we have made will alleviate the reader's confusion about this issue.
Question 3: In the result section, there is no reference to PredictRNA (taken probably form [9]). The nature of this method is not touched too. The comparison with circDeep and with its different extensions and combinations is done in the text, but it is very brief.
Response: Thank you very much for raising this issue, we found that we didn't describe it clearly in the manuscript. In fact, in the beginning, we are only going to compare with models that just input with sequences feature, i.e. the ACNN-BLSTM descriptor in Circdeep, but this classifier also does not use full-length raw RNA sequences as input, and the number of models involved in this comparison is on the low side. So, we added the PredictRNA model and the entire Circdeep model and also incorporated the other two types of input features from the Circdeep model into our model, which also improved the final performance of our model. It is for this reason that the initial description changes were not complete, which caused the reading to be uncomfortable. Thank you very much for raising this question. We have revised the description in result 3.4 to make it more logical and complete.
Question 4: It is not clear why the authors present results in a less detailed form than in previous section 3.3. (sensitivity, specificity, precision).
Response: Thank you very much for this question. Among the various metrics, ACC is the most straightforward metric to evaluate the model, and both MCC and F1 Score are comprehensive metrics integrated by multiple simple metrics (as shown in the Materials and Methods section 2.7) and were chosen by the several articles[9, 13] to validate the model effectiveness as well as for comparison, therefore, in our article, we also choose the three metrics for model comparison. And in the model effectiveness validation part (Result section 3.3), we show all the effectiveness metrics we used for testing the model, including the ROC curve and the corresponding AUC. We aim to show the effectiveness and comprehensiveness of the model more directly under the display of multiple metrics. Thank you again for your interest in this issue, we also hope that the reader can directly see the more detailed information related to the model by exhibiting comprehensive metrics and the training process of the model.
Question 5: In the motif discovery part, I would appreciate a better explanation of the relationship between instances (70nt) and high attention sequences (8-40nt).
Response: I’m very happy that you are interested in this very interesting issue. During the study, we tried different instances length settings for training, and in the results for different instance lengths setting, we found that the length distribution of the attention sequence given by the model is more consistent when its effectiveness is more consistent and its mean value is not relevant to the chosen instance's length. Like in this manuscript, the length of instance is 70nt, while the average length of attention sequence is 40nt. It shows that the extraction mechanism is also effective when the length of the key sequence is smaller than the length of the instance. To be specific, if the key sequence exists in multiple instances that span the region, the multiple instances are all given a higher attention score, so that the key sequence is more prominently displayed in the overlapping region of these instances after their superposition according to the sliding rules. The length of attention sequences is relevant to the chosen threshold for attention score, and considering the mechanism described here, we believe that the threshold is more reasonable when the length of most key sequences is lower than the length of instance. And currently, we are still working on the related issues and including the subsequent enrichment of motifs, since we think the model is interesting and hope to further investigate and experimentally validate it. Thank you again for your interest in this issue and we hope to have more interesting results in the future.
Round 2
Reviewer 2 Report
My previous comments mainly concerned presentational issues. The authors answered all of them somehow. Although I am not completely satiffied with the quality and extent of the changes, I leave other possible improvements to the standard language quality management of the Genes journal.